# Maternal Thyroid Disease and the Risk of Childhood Cancer in the Offspring

**DOI:** 10.3390/cancers13215409

**Published:** 2021-10-28

**Authors:** Laura K. Seppälä, Laura-Maria Madanat-Harjuoja, Maarit K. Leinonen, Mitja Lääperi, Kim Vettenranta

**Affiliations:** 1Pediatrics, Helsinki University Hospital, 00290 Helsinki, Finland; kim.vettenranta@hus.fi; 2Faculty of Medicine, University of Helsinki, 00290 Helsinki, Finland; 3Institute for Statistical and Epidemiological Research, Finnish Cancer Registry, 00130 Helsinki, Finland; laura.madanat@cancer.fi; 4Dana-Farber/Boston Children’s Cancer and Blood Disorders Center, Boston, MA 02115, USA; 5Data and Analytics Unit, Information Services Department, Finnish Institute for Health and Welfare, 00271 Helsinki, Finland; maarit.leinonen@thl.fi; 6Pediatric Research Center, Children’s Hospital, Helsinki University Hospital, 00290 Helsinki, Finland; mitja@laaperi.com; 7Faculty of Medicine, University of Helsinki and the Finnish Red Cross Blood Service, 00290 Helsinki, Finland

**Keywords:** maternal thyroid disease, thyroid hormone, childhood cancer, adolescent cancer, cancer risk, case-control, registry-based

## Abstract

**Simple Summary:**

Maternal thyroid disease, especially hypothyroidism, is known to affect pregnancy and its outcome. We evaluated the risk of childhood cancer in the offspring following exposure to maternal thyroid disease in a case-control setting using registry data. In our study, maternal hypothyroidism was associated with an increased risk of lymphoma in the offspring. The association remained stable when possible familial cancers were excluded.

**Abstract:**

Maternal thyroid disease, especially hypothyroidism, affects pregnancy and its outcome. In-utero exposure to autoimmune thyroid disease has been reported to associate with childhood ALL in the offspring. We evaluated the risk of childhood cancer in the offspring following exposure to maternal thyroid disease in a case-control setting using registry data. All patients with their first cancer diagnosis below the age of 20 years were identified from the Finnish Cancer Registry (*n* = 2037) and matched for sex and birth year at a 1:5 ratio to population controls identified from the Medical Birth Registry (*n* = 10,185). We collected national information on maternal thyroid disease from the Medical Birth Registry, Care Register for Health Care, Register for Reimbursed Drug Purchases and Register of Special Reimbursements. We used conditional logistic regression to analyze childhood cancer risk in the offspring. The adjusted OR for any childhood cancer was 1.41 (95%, CI 1.00–2.00) comparing the offspring of mothers with hypothyroidism and those with normal thyroid function. The risk of lymphomas was increased (adjusted OR for maternal hypothyroidism 3.66, 95%, CI 1.29–10.38). The results remained stable when mothers with cancer history were excluded from the analyses. Maternal hypothyroidism appears to be associated with an increased risk for childhood lymphoma in the offspring. The association exists even after excluding possible familial cancers.

## 1. Introduction

In-utero exposure to maternal disease and its medication has been recognized as potential risk factors for the development of childhood cancer in the offspring. Established risk factors for childhood cancer are sparse with only 10% of the cases being associated with recognized, hereditary predisposition syndromes [1], although this proportion can be an underestimate [2]. Other possible, previously introduced risk factors, include both high and low birth weight [3,4,5,6], preterm birth [7,8,9,10], maternal diabetes [11,12], and some environmental factors, i.e., pesticides [13]. Population-based case-control studies report no association between maternal autoimmune disease other than thyroid and childhood cancer in the offspring [14,15,16,17].

Maternal thyroid disease is known to affect the pregnancy and its outcome in several ways. Maternal hypothyroidism is associated with gestational diabetes, hypertension, and pre-eclampsia [18]. Preterm birth, large for gestational age neonates (LGA), small for gestational age (SGA), and neonatal intensive care are more common among the offspring of mothers with hypo- and hyperthyroidism compared to those of euthyroid mothers [19]. Outcomes such as abortions, intrauterine deaths, and major congenital malformations have been associated with exposure to maternal thyroid diseases, especially untreated [20,21,22].

Maternal thyroid disease, particularly with autoimmune etiology, has been associated with childhood acute lymphoblastic leukemia (ALL) in the offspring [23], while two other studies found no association [24,25]. In adults, nodular thyroid disease is associated with an increased thyroid cancer risk [26]. Hereditary forms of thyroid cancer have also been well described [27]. Maternal thyroid medication and its association with childhood cancer in the offspring has not been previously studied using registry data. A few studies based on self-reported data with inconclusive results for maternal thyroid hormone use and cancer in the offspring do exist [28,29].

Our aim was to evaluate the risk of childhood cancer in the offspring following an in-utero exposure to maternal thyroid disease and its medication in a population-based, registry-derived, case-control study design.

## 2. Materials and Methods

Permanent residents of Finland are given a unique personal identity code. They are covered by the Finnish National Health Insurance and eligible for reimbursement for the cost of prescription drugs. This code allows for the linkage of information from the health and vital statistics registries.

The Finnish Cancer Registry (FCR) started the systematic, nationwide registration of cancer in 1953, and also includes data on cancer treatments and cause of death. The FCR has a 95% coverage for all cancers [30], and 92% for childhood solid tumors and 97% for leukemia [31].

The Finnish Medical Birth Registry (MBR), run by the Finnish Institute for Health and Welfare (THL), contains data on all mothers who have delivered a child in Finland since 1987. Information on the obstetric and neonatal outcomes is available up to 7 days after delivery or at discharge. The quality of the register is considered good or satisfactory (depending on the variable) with less than 1% of births missing and a more than 95% agreement with information in the medical records [32,33].

The Register of Reimbursed Drug Purchases is maintained by the Social Insurance Institute of Finland (Kela) and retains data on all prescription drugs reimbursed since 1993. The registry includes personal information on the Anatomic Therapeutic Chemical (ATC) code of the drug, date of purchase, package size, drug cost, and refund category.

The Register of Special Reimbursements is also maintained by Kela and contains information on the special reimbursement codes since 1986 as well as on the lower and higher special drug reimbursement codes. The Care Register for Health Care (HILMO) contains information on patients, hospital admissions and discharges, diagnoses, and treatment given in the secondary and tertiary health care. It has been maintained by THL since 1969.

The data from all these registries, except HILMO, were collected by the THL in its Drugs and Pregnancy-project to evaluate the maternal use of medication during pregnancy and its outcome [34]. The data have been collected from 1996 onwards and updated by THL. In this study we used this database to collect the data of maternal thyroid disease and its medication from years 1996 to 2014.

The research permits for this study were obtained from the THL (THL/252/5.05.00/2016), Kela (15/52272016) and Helsinki University Hospital. No ethical board review was required with this study being fully registry-based and no study participants being contacted.

### 2.1. Study Population

We identified all individuals in the FCR with their first cancer diagnosis below the age of 20 years for the years 1996–2014 (*n* = 2037). Then, we identified five population-based controls matched for birth year and sex for each case (*n* = 10,185) from the MBR. Due to missing data on birth weight, eight cases and 82 controls were omitted. Maternal cancer is known to be involved with several hereditary predisposition syndromes [2,27]. Hypo- and hyperthyroidism have also been shown to be common late effects of cancer therapy in childhood or young adulthood [35,36,37]. Thus, we also identified all mothers with any cancer diagnosis by the age of 50 years in 1953–2014 (*n* = 344). The descriptive characteristics of the cases and controls are given in Table 1.

### 2.2. Exposure Definition and Classification

For our study, the medical information on the thyroid disease of the mother recorded at any time before the delivery was obtained from the HILMO, MBR, Register of Reimbursed Drug Purchases and Register of Medical Special Reimbursements via its Drugs and Pregnancy database (Figure 1). The International Statistical Classification of Disease and Related Health Problems (ICD) versions ICD-9 and -10 and the special reimbursement code 104 were employed.

Data on the drugs used to treat the thyroid disease and purchased three months prior to conception and/or during pregnancy were obtained from the Register of Reimbursed Drug Purchases. The drugs were identified from the registry using the fifth level ATC codes and grouped as follows: glucocorticoids (H02AB), thyroid hormones (H03AA), antithyroid preparations (H03B) and iodine therapy (H03C), the last of which was later omitted in the absence of purchases.

Combining data on the diagnoses and reimbursed drug purchases, the following mutually exclusive maternal thyroid disease subgroups were formed: hypothyroidism (ICD-9: 243, 244; ICD-10: E03; reimbursement code 104; or ATC group H03AA), hyperthyroidism (ICD-9: 242; ICD-10: E05; or ATC group H03B) and other thyroid disease (ICD-9: 226, 240-241, 245, 246; or ICD-10: E00-02, E04, E06, E07, D34).

The date of conception was calculated as the date of delivery minus gestational age at birth in days based on an ultrasound or best clinical estimate in the absence of the former as registered in the MBR. The birth weight was categorized as small (SGA), appropriate (AGA) or large for gestational age (LGA). SGA was defined as a birthweight under −2 SD and LGA as that over +2 SD of the standard, population-based growth-curves [38].

### 2.3. Cancer Definition and Classification

We defined cancer as a malignant neoplasm but also included benign or borderline tumors of the central nervous system (CNS). The FCR uses the International Classification of Childhood Cancer: with morphology (ICD-O-3), and with morphology and site (ICCC3) (codes 011 for ALL, 011–015 for all leukemias, 021–025 for lymphomas, 031–036 for CNS tumors and 037–122 for other cancers) [39].

We identified a total of 649 (31.9% of all cancer cases) leukemias, of which 511 (25.2%) were ALL. We also identified 149 (7.2%) lymphomas, 484 (23.9%) CNS tumors, and 743 (36.6%) solid tumors other than CNS. 4 (2.0%) tumors were classified as undefined.

For mothers, all malignant neoplasms and benign or borderline CNS tumors diagnosed before age 50 were extracted from the FCR for the years 1953–2014. We identified 344 cancer cases among the mothers. Of these, 309 (90% of all maternal cancers) were diagnosed at least 40 weeks (280 days) before the child’s birthday. In total, 39 cancers (10%) were diagnosed during pregnancy or after the birth.

### 2.4. Statistical Analysis

We evaluated the association between the maternal thyroid disease, its medication and the risk of childhood cancer in the offspring. We used conditional logistic regression to estimate the odds ratios (OR) and 95% confidence intervals (CI) for childhood cancer among each thyroid disease subgroup as follows: hypothyroidism, hyperthyroidism, and other thyroid disease, compared to the offspring of mothers without any thyroid disease.

Maternal age is known to associate with increased risk for childhood cancer [40,41]. Older mothers have also more chronic diseases, such as hypothyroidism [34]. The role of parity is not as clear—there are reports of increased risk for childhood cancer increasing with parity [42,43], but also opposite results have been presented [44,45]. Parity increases with maternal age. Maternal smoking has previously shown to associate with childhood cancer in the offspring [43,46]. It is also a good proxy for socioeconomic status in Finland. These were kept as possible confounding factors in our study.

The crude model was repeated adjusting for the maternal age (<25, 25–29, ≥30 years), parity (primiparous, multiparous) and maternal smoking status during pregnancy (yes/no). The information on smoking was missing for 58 cases and 252 controls and thus omitted from the adjusted models.

As maternal cancer is associated with cancer in the offspring through possible familial cancer predisposition syndromes, and both hypo- and hyperthyroidism are common late effects of cancer in childhood or early adulthood, we ran another adjusted model including also maternal cancer as a covariate.

Both low and high birth weight have been associated with childhood cancer risk [5,6], and mothers with thyroid disease are known to have both SGA and LGA babies, making birth weight a potential mediator of the relationship between the maternal thyroid disease and offspring cancer risk. To account for this, we conducted a sensitivity analysis adjusting for the birth weight categorized into three groups: <2500 g, 2500–4500 g, and >4500 g. We also performed subgroup analyses with selected childhood cancer subtypes.

We further analyzed the data on thyroid hormone medication by stratifying the drug purchases into two groups, i.e., those during the 3 months before pregnancy and/or during the 1st trimester (exposure in early pregnancy yes/no) and those during the 2nd and/or 3rd trimester(s) (exposure in late pregnancy yes/no). Women on medication throughout the pregnancy contributed to both categories. The statistical analyses were performed with the StataMP version 14 (StataCorp LLC, College Station, TX, USA).

## 3. Results

In our study, we identified a total of 200 (1.6% of all) mothers with hypothyroidism. We identified eight mothers with hypothyroidism diagnosis but no medication, 96 mothers with both diagnosis and thyroid hormone purchases, 92 mothers with no hypothyroidism diagnosis, but no thyroid hormone purchases and four mothers with both hypo-and hyperthyroidism diagnosis and thyroid hormone medication. A total of 14 (0.1%) mothers were with hyperthyroidism during pregnancy. Eight (0.07%) mothers had other thyroid diseases (Table 1).

The baseline adjusted OR for any childhood cancer after exposure to maternal hypothyroidism was 1.41 (95% CI 1.00–2.00) and for hyperthyroidism 0.41 with 95% CI 0.05–3.14 (Table 2).

The OR for childhood lymphoma in the offspring was significantly increased following exposure to maternal hypothyroidism (baseline adjusted OR 3.66, 95% CI 1.29–10.38). Of the seven cases, one was classified as Hodgkin’s, six as non-Hodgkin’s, and two as Burkitt lymphoma. No other significant associations with maternal thyroid disease and childhood cancer subtypes (leukemia, CNS tumors, solid tumors other than CNS) in the offspring were found (Table 3). We performed sensitivity analyses by further adjusting our model for the birth weight (categorized), but the results did not differ from the model presented above adjusting for the maternal age, parity, and smoking status in pregnancy (Table A1).

No difference in the OR for maternal thyroid hormone use and childhood cancer was detected over time with a baseline adjusted OR of 1.50 (95% CI 1.01–2.20) for early, and 1.50 for late pregnancy (95% CI of 1.05–2.14). For glucocorticoids the respective ORs were 0.92 (95% CI 0.49–1.71) and 1.30 (95% CI 0.67–2.54). The sample size was too small to analyze the antithyroid medication by trimester.

## 4. Discussion

By harnessing population-based, nationwide data from various national registries with multiple sources of information on exposure, we were able to study maternal thyroid disease, its medication, and the risk of childhood cancer among the offspring.

In our case-control study, the prevalence of thyroid disorders among the mothers before or during pregnancy was similar to that previously reported by a Danish group [47], and slightly lower compared to the Finnish birth cohort studies [18,19].

Maternal thyroid disease, especially hypothyroidism, has been associated with several perinatal problems: preterm birth, LGA, SGA, neonatal intensive care as well abortions, intrauterine deaths, and major congenital malformations [18,48,49]. Therefore, it is easy to postulate maternal thyroid disorder also affecting the development of childhood cancer among the offspring. Our results emerge supportive of this hypothesis with medicated maternal hypothyroidism being associated with childhood lymphoma. Unfortunately, there is no strong biological explanation for this association yet. We know that thyroid hormone passes the placenta and is necessary for the normal fetal growth and development in the uterus [21]. Altering thyroid hormone levels may have an impact on cancer process as well.

Previous, interview-based studies on maternal thyroid hormone therapy found a non-significant association with childhood neuroblastoma [28] and a possible protective impact on infant leukemias [29]. We were unable to investigate these cancer subtypes due to low numbers, but any childhood leukemia showed a minimally increased, non-significant risk (Table 3).

In Finland, thyroid hormone is a very cheap drug and therefore some of the purchases may not get registered. In our study we also lacked information about primary outpatient care diagnoses, offering possible risk for misclassification about hypothyroidism. Yet, our mutually exclusive group formation was able to take into account both the disease and the medication when evaluating the exposure of maternal hypothyroidism.

In adults, there are some rare cases of Hashimoto’s thyroiditis having developed a lymphoma in the thyroid [50], and sometimes it is difficult to differentiate Hashimoto’s from lymphoma histologically [51]. An increased risk for lymphomas among the offspring of mothers with thyroid disease has been reported previously for non-Hodgkin’s lymphoma [15], but discordant results have also been published [24]. Our result should be confirmed using a larger dataset to be able to categorize the lymphomas and tease out the possible underlying mechanisms.

The strengths of our study are the population-based setting and possibility to extract information on the exposure from multiple, nationwide registries, thus avoiding a recall and selection bias. Studying a rare outcome and relatively rare exposure, our study was hampered by a small sample size. We lacked power to explore the risk of all the subgroups of pediatric cancer reliably.

To our knowledge, this is the first registry-based study reporting an association between maternal hypothyroidism and risk of childhood lymphoma among the offspring. Using our data on maternal cancer we were also able to evaluate the risk for childhood cancer in the offspring independently from possible, familial cancer predisposition syndromes. Our nationwide, population-based registry data thus gave a reliable estimate on the association between a maternal thyroid disease and the increased risk of childhood cancer among the offspring.

## 5. Conclusions

In our nationwide, population-based registry study with verified data on outcome, we found maternal hypothyroidism to be associated with an increased risk for childhood lymphoma and reflected to the results of any cancer. The results remained stable even after adjusting for maternal cancer. The results need further confirmation with a larger dataset.

## Figures and Tables

**Figure 1 cancers-13-05409-f001:**
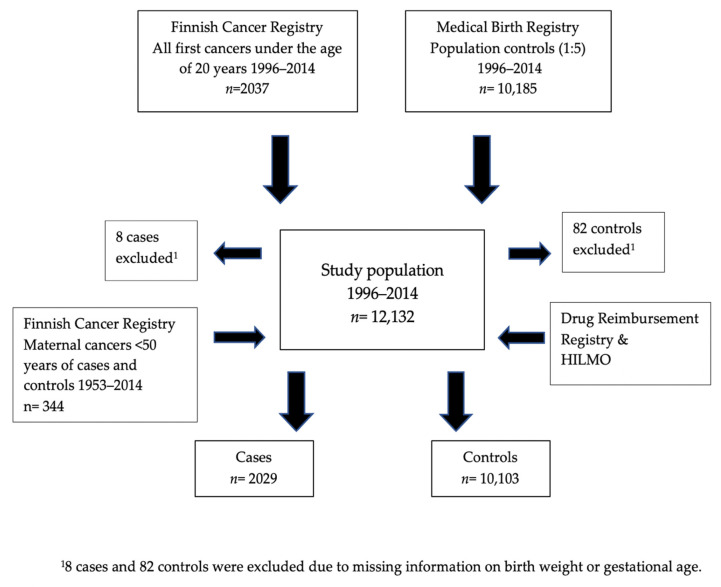
The data compilation for the years 1996–2014.

**Table 1 cancers-13-05409-t001:** The descriptive characteristics of the cases and controls, years 1996–2014.

	Number of Cases = 2029 (%)	Number of Controls = 10,103 (%)	Fisher’s Exact *p*-Value
Maternal characteristics			
Maternal age			0.35
<25	355 (17.5)	1872 (18.5)	
25–29	639 (31.5)	3246 (32.1)	
≥30	1035 (51.0)	4985 (49.3)	
Parity			0.32
Primiparous	847 (41.7)	4098 (40.6)	
Multiparous	1182 (58.3)	6005 (59.4)	
Maternal smoking			0.77
Yes	291 (14.3)	1447 (14.3)	
No	1680 (82.8)	8404 (83.2)	
Unknown	58 (2.9)	252 (2.5)	
Maternal thyroid disease subgroup			
Hypothyroidism	45 (2.2)	155 (1.5)	0.04
Hyperthyroidism	1 (0.05)	13 (0.1)	0.49
Other thyroid disease	3 (0.1)	5 (0.05)	0.14
Maternal cancer	64 (3.2)	280 (2.8)	0.34
Offspring characteristics			
Sex			0.96
Male	1092 (53.8)	5431 (53.8)	
Female	937 (46.2)	4672 (46.2)	
Gestational age			0.004
Full-term (≥37 weeks)	1888 (93.1)	9566 (94.7)	
Preterm (<37 weeks)	141 (6.9)	537 (5.3)	
Multiple pregnancy			0.68
Yes	67 (3.3)	319 (3.2)	
No	1962 (96.7)	9784 (96.8)	
Delivery type			0.13
Vaginal	1906 (93.9)	9571 (94.7)	
Caesarian section	123 (6.1)	526 (5.2)	
Unknown	0 (0)	6 (0.06)	
Birth weight			<0.001
<2500 g	110 (5.4)	395 (3.9)	
2500–4500 g	1833 (90.3)	9412 (93.2)	
>4500 g	86 (4.2)	296 (2.9)	

**Table 2 cancers-13-05409-t002:** Maternal thyroid disease and the risk of childhood cancer in the offspring. Comparison to those of mothers without thyroid disease.

	Number of Cases 2029	Number of Controls 10,103	Adj. OR ^1^	95% CI	Adj.OR ^2^	95% CI
Maternal hypothyroidism	45 (2.2)	155 (1.5)	1.41	1.00–2.00	1.42	1.01–2.01
Maternal hyperthyroidism	1 (0.05)	13 (0.1)	0.41	0.05–3.14	0.41	0.05–3.14
Other thyroid disease	3 (0.1)	5 (0.05)	3.70	0.83–16.54	3.70	0.83–16.54

^1^ Matched model, adjusted for maternal age (categorized), parity, and smoking status. ^2^ Matched model ^1^ adjusting also for maternal cancer.

**Table 3 cancers-13-05409-t003:** Maternal thyroid disease and the risk for selected cancer subtypes in the offspring. Comparison to those of mothers with no thyroid disease.

	Number of Cases	Adj. OR ^1^	95% CI	Adj. OR ^2^	95% CI
ALL	8	0.93	0.41–2.13	0.96	0.42–2.21
Any leukemia	13	1.24	0.64–2.39	1.27	0.66–2.45
Lymphoma	7	3.66	1.29–10.38	3.66	1.29–10.38
CNS	11	1.46	0.72–3.00	1.47	0.72–3.00
Other solid tumor	10	1.19	0.66–2.15	1.19	0.66–2.15

^1^ Matched model adjusted for maternal age (categorized), parity and smoking status. ^2^ Matched model ^1^ adjusting also for maternal cancer.

## Data Availability

The source data that support the findings of this study are not publicly available. According to the current national data protection legislation, the permission to obtain the data must be applied for from Findata (https://www.findata.fi/en/, accessed on 22 September 2021).

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
