# Peer review of "Maternal Thyroid Disease and the Risk of Childhood Cancer in the Offspring"

_cancers, 2021, doi:10.3390/cancers13215409_

Round 1
Reviewer 1 Report
The study is an important epidemiological analysis about maternal thyroid disease and the risk of childhood cancer in the offspring. The authors studied maternal thyroid diseases, its medication and the risk of different pediatric malignant tumors. The design is appropiated and the results are based on various national population-based registries and nationwide data. The authors excluded other factors like posible familial cancer predisposition syndromes and maternal cancer.
Although the results need further confirmation with a larger sample, the conclusion of association between maternal hypothyroidism and the risk of childhood lymphoma among the offspring is relevant. It’s an important contribution for improving the identification of potential risk factors for the development of pediatric cáncer, especially the in-utero exposure to maternal diseases and medications
However, in the results I found a mistake in the description of lymphoma cases that should be solved. The authors describe only 6 cases (1 Hodgkin’s, 4 Non-Hodgkin and 1 Burkitt lymphoma), instead of 7 as they wrote. Also a change in the classification should be included as Burkitt lymphoma is a type of Non-Hodgkin lymphoma. This information is important for the interpretation of the results and the conclusions.
Author Response
We thank the reviewer for an excellent comment. There was a typo in the numbers of lymphomas, and the sentence in the manuscript needed clarification. One of the lymphoma cases was classified as Hodgkin´s, other six as non-Hodgkins, and two of them were Burkitt lymphomas.
The sentence on page 6, line 20 is written now: "Of the seven lymphoma cases one was classified as Hodgkin´s, six as non-Hodgkin, and of them, two as Burkitt lymphoma."
Reviewer 2 Report
This paper presents an assessment of the association between maternal thyroid disease (defined by diagnosis codes or medication dispensings) and childhood cancer in offspring using a reliable population-based data source (the Finnish healthcare registers). The study design and approach are appropriate given the rare outcome (childhood cancer) and the relatively rare exposure (maternal thyroid disease during pregnancy). I have several comments for the authors to consider:
- Line 84-85: I’m unsure about what this sentence means: “The registry includes personal information on ATC code…etc.” – what does “personal information” mean here?
- Line 92: A few grammatical errors were noted in this sentence – suggested edits are in red: “The data from all these registries, except HILMO, were collected by THL…”
- Please note throughout the manuscript that the word “data” should be treated as plural (i.e., “the data were…”; “the data are…”; etc.)
- Figure 1 and Table 1 should be presented as part of the Results (instead of the Methods) since these are in fact results.
- Figure 1: I find this schematic confusing. It seems that the boxes with the exclusions should perhaps be coming out of the middle box (“Study population 1996-2014”) to make the numbers make sense. Also, I’m not sure how the “Finish Cancer Registry Maternal cancers…” box and the “Drug Reimbursement Registry & HILMO” box are meant to relate to the other boxes in this schematic.
- Lines 156-159: Please describe how the covariates were selected for the adjusted model.
- Table 1 and Table 2: Do the Finnish registers have restrictions on presenting numbers with small cell counts? There are several cells with 5 or less cases presented in these tables.
- What is the proportion of mothers with thyroid disease with a dispensing for a thyroid medication? Were there any mothers with a dispensing for a thyroid medication who did not have a diagnosis code for thyroid disease?
- Is there a proposed mechanism of action for how exposure to maternal thyroid disease may predispose offspring to childhood cancer?
- Lines 212-213: Please list the “perinatal problems” that are associated with maternal thyroid disease.
Author Response
We thank the reviewer for important questions and comments. Please find our point by point response below.
Line 84-85: I’m unsure about what this sentence means: “The registry includes personal information on ATC code…etc.” – what does “personal information” mean here?
Response to the reviewer:
Thank you for the comment. This means that in the registry, medication purchases are registered through social security code for every person by their Anatomical Therapeutic Chemical- code. The data is pseudonymized but this way of registration allows us to study medication and combine the information with other data as we did in this study.
Line 92: A few grammatical errors were noted in this sentence – suggested edits are in red: “The data from all these registries, except HILMO, were collected by THL…”
Response to reviewer:
Thank you for this comment. We have corrected the errors mentioned above to page 2, line 48. It says now: "The data of all these registries, except HILMO, were collected by THL in its Drugs and Pregnancy-project, to evaluate the maternal use of medication during pregnancy and its outcome [34].
- Please note throughout the manuscript that the word “data” should be treated as plural (i.e., “the data were…”; “the data are…”; etc.)
Response to reviewer:
We thank the reviewer for this important notification. The use of word "data" has been edited throughout the manuscript.
- Figure 1 and Table 1 should be presented as part of the Results (instead of the Methods) since these are in fact results.
Response to reviewer:We thank the reviewer of this comment. We agree that on table 1, there are some results presented. There is also description of the study population. In our point of view, Figure explains the data formation and it is important to mention in Methods. We have moved Table 1 and Figure closer to results and added a mentioning about Table 1 to the end of the first chapter of the results.
Figure 1: I find this schematic confusing. It seems that the boxes with the exclusions should perhaps be coming out of the middle box (“Study population 1996-2014”) to make the numbers make sense. Also, I’m not sure how the “Finish Cancer Registry Maternal cancers…” box and the “Drug Reimbursement Registry & HILMO” box are meant to relate to the other boxes in this schematic.
Response to reviewer:
We thank the reviewer for this important notification.
We have modified Figure 1 to clarify the relations and added some arrows to the boxes mentioned in reviewers comment to give more clear picture of the data formation.
- Lines 156-159: Please describe how the covariates were selected for the adjusted model.
Response to the Reviewer:
We thank the reviewer for this important comment. We have added a text to page 4, lines 10-16, saying: "Maternal age is known to associate with increased risk for childhood cancer[40,41]. Older mothers have also more chronic diseases, such as hypothyroidism[34]. The role of parity is not as clear- there are reports of increased risk for childhood cancer increasing with parity[42,43], but also opposite results have been presented[44,45]. Parity increases with maternal age. Maternal smoking has previously shown to associate with childhood cancer in the offspring[43,46]. It is also a good proxy for socioeconomic status in Finland. These were kept as possible confounding factors in our study."
- Table 1 and Table 2: Do the Finnish registers have restrictions on presenting numbers with small cell counts? There are several cells with 5 or less cases presented in these tables.
Response to reviewer:
Thank you for this excellent question. There is no legislation in Finland regarding the reporting low cell counts in registry data. At this point there is a lot of variation in practices how these are reported. When studying a rare outcome with a possibly rare exposure, low cell counts are inevitable. In our study we decided to report also low numbers to give a better description of thyroid disease situation among mothers in Finland.
What is the proportion of mothers with thyroid disease with a dispensing for a thyroid medication? Were there any mothers with a dispensing for a thyroid medication who did not have a diagnosis code for thyroid disease?
Response to reviewer:
Thank you for this important question. In Finland, thyroid hormone is a very cheap drug. In our study, only reimbursed drugs were registered. Hypothyroidism is also often treated in primary health care, and since we only had diagnoses from hospitalisations, birth registry and specialised outpatient care, there might some misclassification or underestimate in our data.
We added this desription to results to page 8, lines 13-16, saying: "We identified 8 mothers with hypothyroidism diagnosis but no medication, 96 mothers with both diagnosis and thyroid hormone purchases, 92 mothers with no hypothyroidism diagnosis, but no thyroid hormone purchases and four mothers with both hypo-and hyperthyroidism diagnosis and thyroid hormone medication."
We added this comment to the discussion, to page 9, lines 24-29, saying: "In Finland, thyroid hormone is very cheap drug and therefore some of the purchases may not get registered. In our study we also lacked information about primary outpatient care diagnoses, offering possible risk for misclassification about hypothyroidism. Yet, our mutually exclusive group formation was able to take into account both the disease and the medication when evaluating the exposure of maternal hypothyroidism."
- Is there a proposed mechanism of action for how exposure to maternal thyroid disease may predispose offspring to childhood cancer?
Response to reviewer:
Thank you for this excellent question.There are not many studies reporting an association with hypothyroidism and childhood cancer. Unfortunately, there is no strong biological explanation for this association yet. We know, that thyroid hormone passes the placenta and is necessary for the normal growth and development in the uterus. Altering thyroid hormone levels may have an impact on cancer process as well.
We added a sentence to page 9, lines 19-22, saying "Unfortunately, there is no strong biological explanation for this association yet. We know, that thyroid hormone passes the placenta and is necessary for the normal fetal growth and development in the uterus[21]. Altering thyroid hormone levels may have an impact on cancer process as well.
Lines 212-213: Please list the “perinatal problems” that are associated with maternal thyroid disease.
Response to reviewer:
Thank you for this comment. We have added a list of perinatal problems listed in the introduction section also to the discussion, page 9, lines 15-17, saying : "
Maternal thyroid disease, especially hypothyroidism, has been associated with several perinatal problems : preterm birth, LGA,SGA, neonatal intensive care as well as abortions, intrauterine deaths and major congenital malformations ."
Reviewer 3 Report
The paper is well written and intersting.
I have only one question
The authors attribute to hypothyroidism a possible relation to cancer in the offspring; however it is not clear whether those mothers with hypothyroidism were on therapy or not; in other words, is it possible that cancer is relatad to medication, rather than hypothyroidism itself? The results do not report such detail
Author Response
We thank the reviewer for this very important question. In Finland, thyroid hormone is very cheap drug and since in this study we had only access to reimbursed medication purchases instead of e.g. prescriptions and access to specialised outpatient care diagnoses instead of primary care data, we had 92 mothers with thyroid hormone purchases but no hypothyroidism diagnosis and 8 mothers with hypothyroidism and no thyroid hormone purchases.
To build the most comprehensive and yet mutually exclusive exposure groups we combined all the information about maternal diagnoses, reimbursement codes and thyroid medication purchases. It is also noteworthy that thyroid medication in used mostly to treat hypothyroidism, so differentiating hypothyroidism from thyroid hormone medication is not simple.
We edited the manuscript from page 6, lines 13-16 by saying: "We identified 8 mothers with hypothyroidism diagnosis but no medication, 96 mothers with both diagnosis and thyroid hormone purchases, 92 mothers with no hypothyroidism diagnosis, but no thyroid hormone purchases and four mothers with both hypo-and hyperthyroidism diagnosis and thyroid hormone medication"
On page 7, lines 23-28 we wrote: "
In Finland, thyroid hormone is very cheap drug and therefore some of the purchases may not get registered. In our study we also lacked information about primary outpatient care diagnoses, offering possible risk for misclassification about hypothyroidism. Yet, our mutually exclusive group formation was able to take into account both the disease and the medication when evaluating the exposure of maternal hypothyroidism."